# OpenReview forum: "Jailbreak LEGO: A Compositional Benchmark for Red-Teaming LLMs"
_ICLR.cc/2026/Conference — ICLR 2026 Conference Withdrawn Submission_

### Official Review · Reviewer_vaVW · 2025-10-28

**Soundness:** 3
**Presentation:** 3
**Contribution:** 1
**Rating:** 2
**Confidence:** 5

**Summary:**

The paper proposes a jailbreak benchmarking framework aimed at capturing the compositional nature of jailbreak attacks. It extracts modular and composable strategy components from existing attacks, and categorizes them according to their transformation behavior, thus creating rules for their composition. The paper benchmarks 16 attacks (turned into 26 reusable primitives) and their compositions on 8 LLMs. The compositional attacks created by the approach achieve high ASRs for several standard LLMs.

**Strengths:**

- This work creates a bank of reusable attack primitives for future work to build on. It is also quite timely given the increasing prevalence of LLMs.
- The paper presents an unified evaluation of 16 attacks from literature, providing the opportunity to compare their effectiveness. New attacks created from the composition of these attacks also prove to be effective, reaching 91% ASR with Claude-3.7 as the target LLM.
- The conversion of jailbreak attacks into string triples is also logical and allows for an unified representation across attacks.

**Weaknesses:**

- The main weakness of this work is its weak differentiation from earlier work on compositional jailbreak attacks - namely [1]. To start with, attacks in both h4rm3l and Jailbreak LEGO are compositional, so I did not find the argument starting on line 144 compelling. While the set of primitives covered in this work is more extensive (26 compared to 22 in h4rm3l), it is also more limited along several dimensions, which I list below. Therefore, my opinion is that this work is incremental in the context of h4rm3l.
- h4rm3l supported the composition of 2 or more attack primitives, and also allowed for parameterized primitives that could represent different variations of the same attack. h4rm3l was also a complete framework, since it had primitives that allowed for the representation of all possible string transformations inside it. In contrast, Jailbreak LEGO uses rigid strategy components, and only explores the composition of 2 primitives. h4rm3l also found that composing more primitives generally led to higher ASRs.
- h4rm3l had an automatic composition generation component that allowed an LLM to automatically discover new effective attack compositions dependent on a target LLM. In contrast, Jailbreak LEGO tries out all possible attack compositions and presents no such approach, which does not scale as the space of primitives increases. The “guided search” strategy presented simply involves trying out all possible primitive compositions on a subset of the evaluation dataset, which faces the same issues with scaling.
- While the work shows that integrating their jailbreaks with PAIR improves its performance, it does not baseline against doing the same with other existing jailbreak attacks, such as the single strategy components presented in the paper.

[1] h4rm3l: A language for Composable Jailbreak Attack Synthesis. Moussa Koulako Bala Doumbouya, Ananjan Nandi, Gabriel Poesia, Davide Ghilardi, Anna Goldie, Federico Bianchi, Dan Jurafsky, Christopher D. Manning.

**Questions:**

- Do the attack primitives curated by Jailbreak LEGO recover the original reported performance numbers from the respective papers?
- A human agreement analysis for the chosen ASR metric would make the results more convincing. Harmbench-Llama was finetuned for Harmbench, not JBB-Behaviors.

---

### Official Review · Reviewer_qK25 · 2025-10-29

**Soundness:** 3
**Presentation:** 3
**Contribution:** 2
**Rating:** 2
**Confidence:** 4

**Summary:**

This paper proposes Jailbreak LEGO, a compositional benchmark that decomposes existing jailbreak attacks into atomic “strategy components” (Change Prompt, Generate Template, Change Template) and recombines them like LEGO blocks to create new adversarial prompts. By formalizing jailbreaks as operations on structured triples and testing 16 methods across 8 LLMs, the authors show that compositional attacks can uncover unseen vulnerabilities (e.g., 91 % ASR on Claude-3.7) and improve existing frameworks such as PAIR, establishing a systematic and extensible approach for red-teaming large language models.

**Strengths:**

1. Offering a good perspective on reusing and combining natural-language jailbreak attack strategies.
3. Demonstrates significant performance improvements (e.g., 91 % ASR on Claude-3.7) and uncovers previously unseen vulnerabilities.
4. Provides open-sourced code, enhancing reproducibility and transparency for future research.

**Weaknesses:**

1. The comparison with prior work, particularly h4rm3l, feels somewhat overstated. Simply including more referenced papers does not necessarily constitute a substantial contribution, and the claim that the triplet-based design enables “seamless application of any general attack component” lacks a clear definition or empirical justification. Could the authors clarify what “seamless” specifically means in this context? Does it refer to theoretical guarantees of composability?

2. While the paper formalizes jailbreaks as three components, the proposed framework seems to lack diversity. For instance, attacks like GCG [1] do not appear to fit into any of the defined categories, and reasoning-based jailbreaks [2] also seem incompatible. Could the authors clarify whether GCG can be represented within their formulation, and if not, explain why? According to Figure 2, GCG obviously does not align with the triplet transformation process.

3. No defense or mitigation analysis is provided, which substantially limits the practical relevance of the benchmark. While the paper focuses on attack synthesis and expansion of the threat space, it does not explore how current or potential defense mechanisms perform under these compositional attacks. Could the authors include or discuss any preliminary evaluation of defenses under their framework?

4. The paper lacks a clear component ablation analysis, making it difficult to assess which specific strategies or component types contribute most to the observed performance gains.

5. The scope of the work appears somewhat inconsistent: while the authors position the study as a comprehensive benchmark, it focuses exclusively on jailbreak attacks and excludes reasoning-based models such as GPT-o1, DeepSeek-R1, and QwQ. Could the authors clarify why these models were omitted?

6. The component extraction process is manual, which limits scalability and may introduce subjective bias.

7. The evaluation relies heavily on a single automated evaluator (Harmbench-Llama-2-13b), which could bias ASR estimation and borderline classifications. How does this evaluator align with human annotations?

References.

[1]. Zou, A., Wang, Z., Carlini, N., Nasr, M., Kolter, J. Z., & Fredrikson, M. (2023). Universal and transferable adversarial attacks on aligned language models. arXiv preprint arXiv:2307.15043.

[2]. Kuo, M., Zhang, J., Ding, A., Wang, Q., DiValentin, L., Bao, Y., ... & Chen, Y. (2025). H-cot: Hijacking the chain-of-thought safety reasoning mechanism to jailbreak large reasoning models, including openai o1/o3, deepseek-r1, and gemini 2.0 flash thinking. arXiv preprint arXiv:2502.12893.

**Questions:**

See my weaknesses above.

---

### Official Review · Reviewer_zDSc · 2025-11-01

**Soundness:** 3
**Presentation:** 2
**Contribution:** 2
**Rating:** 4
**Confidence:** 4

**Summary:**

This paper presents Jailbreak LEGO for evaluating LLM security. The method decomposes existing jailbreak attacks into strategy components, which are formalized as operations on a (query, template, attack_prompt) triple. These components are categorized into three types based on how they modify this structure. The method then combines these components to generate new, composite attack strategies and uses a guided search algorithm to identify effective combinations.

**Strengths:**

* Originality: This paper introduces a new method that deconstructs existing jailbreak attacks into composable atomic components, which enables the systematic generation of new attack strategies.
* Clarity: The method is well-written with clear formalizations.
* Significance: The work provides a more systematic and scalable approach to LLM red-teaming.

**Weaknesses:**

* The method's foundational claim of decomposing attacks into atomic components is questionable. The granularity of these components appears to be an arbitrary design choice, dictated by the initial set of attacks selected for analysis. There is no theoretical grounding to suggest these components are truly indivisible, which undermines the authors' claim to be a systematic exploration of the strategy space. I think it is more likely an ad-hoc factorization of a biased sample.

* The method operates on the naive assumption that randomly combining these atomic parts will produce coherent new attacks. It lacks any semantic understanding of why certain strategies work, which treats them as interchangeable blocks. This overlooks the high probability that many compositions are nonsensical, redundant, or even self-defeating. This would rely on brute force search to stumble upon success rather than intelligent design.

* The formalization of an attack as a static (query, template, attack_prompt) triple is a significant limitation. This structure is incapable of representing more sophisticated, multi-turn conversational attacks that rely on manipulating conversational history and context. The method is therefore inherently biased towards a narrow class of single-shot, prompt-based attacks, which ignores a potentially large and more dangerous set of interactive vulnerabilities.

**Questions:**

* Regarding Weakness 1, what is the specific set of criteria used to determine that a component is atomic and could not be further subdivided?  have the author considered alternative decomposition taxonomies? how might a different scheme for defining atomicity change the resulting component library and the method's effectiveness?

* Regarding Weakness 2, how does the method handle the composition of components that are logically contradictory or even mutually exclusive? is there a mechanism to resolve such conflicts, or does it rely solely on the final evaluation score to discard them?
* Regarding Weakness 3, by focusing exclusively on single-shot prompts, is it not possible that the framework overlooks critical, emergent vulnerabilities that only manifest in the context of an extended dialogue? could the method be extended to represent and generate multi-turn attacks that require adapting to the model's responses over several turns?

---

### Official Review · Reviewer_rj8E · 2025-11-04

**Soundness:** 2
**Presentation:** 3
**Contribution:** 2
**Rating:** 2
**Confidence:** 5

**Summary:**

This paper proposes Jailbreak LEGO, a framework for composing jailbreak attacks on LLMs by extracting atomic strategy components from existing attacks. The authors formalize jailbreak prompts as structured triples (Query, Template, Attack_prompt) and categorize components into three functional types. They evaluate 16 jailbreak methods on 8 LLMs and explore compositions of these components, claiming to discover novel vulnerabilities. While addressing an important problem, the work has significant weaknesses in its treatment of prior literature, limited compositional exploration, and unclear contributions over existing frameworks.

**Strengths:**

- This work addresses an important and timely issue
- This work attempts to propose a formalism to explore the compositional space of black-box jailbreak attacks
- This work reproduced several previously published attacks and evaluated 8 SOTA LLMs

**Weaknesses:**

- The claim that prior approaches "fall short in capturing the compositional nature of real-world attacks" is inaccurate. [2] showed that the composition of black-box attacks can lead to more successful attacks. [1] proposed a compositional language (h4rm3l) that can represent any black-box attack, and a method to synthesize optimized attacks in that language.
- The terminologies and formalisms adopted are confusing and unnecessarily diverge from prior proposals. In particular:
  - **Line 63**: Why are "jailbreak prompts" represented as the triple (Query, Template, Attack_prompt)?  Aren't these strings resulting from a transformation applied by a black-box attack?
  - How are black-box attacks defined? Why depart from prior formalisms representing attacks as string transformations (and composition thereof)?
- This work does not accurately discuss prior literature regarding the composability of black-box attacks and previously proposed methods that specifically address this issue (see [1, 2, 3]).
- **Line 239** The work claims to enable comprehensive exploration of the compositional space of black-box attacks, but only proposes compounds of size 2.
- The proposed attack synthesis method is not tractable. The enumeration of the combinatorial space of primitives is infeasible.
- The proposed representation doesn't seem to be capable of representing all possible black-box attacks (all computable string transformations). For instance, it is not clear how to implement the following attack primitives in the proposed formalism: Base64, Low-Resource translation, and payload splitting. Prior representations such as [1]  enable primitives that perform arbitrary computation on strings.

**Questions:**

- Why is it necessary to deviate from the composable string-to-string transformation interface proposed by [1]? What are the limitations of [1] that are addressed by the proposed formalism?
- Why is the proposed synthesis method restricted to compounds of size 2? [1] synthesized several attacks composed of more primitives, and empirically showed that ASRs tend to increase with composition complexity.
- What are the limitations of the proposed synthesis method? Is the enumeration of the combinatorial space of primitives tractable? how does this compare to the few-shot bandit-based approach proposed by [1]?
- How are the following jailbreak attack primitves represented in the proposed formalism?
    - Base64 Attack [4]. This falls under the "Mismatched Generalization" category (as defined by [2]). It is a string-to-string transformation that outputs the base-64 encoded version of the input prompt. How does this map to the proposed (Query, Template, Attack_prompt) formalism?
    - Low-resource translation attack [5]. This is a string-to-string transformation that outputs the translated version of the input prompt in a different language (e.g. the authors used Zulu)
    - Payload-splitting attack [6]
- [1] Claimed that h4rm3l can represent all possible black-box attacks, and implemented several primitives from prior literature (appendix B3). How can these primitives be implemented using the proposed (Query, Template, Attack_prompt) formalism?
- **Line 139**: The claim that the proposed formalism broadens the space of covered attacks compared to [1] because more primitives, and more recent ones are enclosed in the present work is flawed. The authors should accurately compare the representation space covered by the proposed formalism and the one introduced in [1]
- **Line 145**. The claim that the proposed framework employs a structured and flexible design setting it appart from prior work is flawed. The representation proposed in [1] is structured and flexible, and can represent the full set of black-box jailbreak attack, which corresponds to the set of all computable string-valued function of strings. The authors should clarify their comparison to prior work [1].
- How are ASRs computed and validated?
- **Line 41**: "the absence of standardized evaluation frameworks has made it difficult to fairly assess and compare their effectiveness." Is this accurate? Standardized datasets like AdvBench, HarmBench, and StrongReject have been helpful for evaluating jailbreak attacks on an equal footing. Additionally the previously proposed domain-specific-language [1] enables a uniform representation for black-box attack primities and arbitrary compositions thereof.



# References
- [1] Doumbouya, M. K. B., Nandi, A., Poesia, G., Ghilardi, D., Goldie, A., Bianchi, F., ... & Manning, C. D. h4rm3l: A Language for Composable Jailbreak Attack Synthesis. In The Thirteenth International Conference on Learning Representations.
- [2] Wei, A., Haghtalab, N., & Steinhardt, J. (2023). Jailbroken: How does llm safety training fail?. Advances in Neural Information Processing Systems, 36, 80079-80110.
- [3] Sharma, M., Tong, M., Mu, J., Wei, J., Kruthoff, J., Goodfriend, S., ... & Perez, E. (2025). Constitutional Classifiers: Defending against Universal Jailbreaks across Thousands of Hours of Red Teaming. CoRR.
- [4] Greshake, K., Abdelnabi, S., Mishra, S., Endres, C., Holz, T., & Fritz, M. (2023, November). Not what you've signed up for: Compromising real-world llm-integrated applications with indirect prompt injection. In -
- [5] Proceedings of the 16th ACM workshop on artificial intelligence and security (pp. 79-90).
Yong, Z. X., Menghini, C., & Bach, S. H. (2023). Low-resource languages jailbreak gpt-4. arXiv preprint arXiv:2310.02446.
- [6] Kang, D., Li, X., Stoica, I., Guestrin, C., Zaharia, M., & Hashimoto, T. (2024, May). Exploiting programmatic behavior of llms: Dual-use through standard security attacks. In 2024 IEEE Security and Privacy Workshops (SPW) (pp. 132-143). IEEE.

---

### Note · Authors · 2025-11-16

I have read and agree with the venue's withdrawal policy on behalf of myself and my co-authors.